# Structured Uncertainty in the Observation Space of Variational Autoencoders

**James Langley**                                                    *jwblangley@gmail.com*
*Department of Computing, Imperial College London, UK*

**Miguel Monteiro**                                          *miguel.monteiro@imperial.ac.uk*
*Department of Computing, Imperial College London, UK*

**Charles Jones**                                            *charles.jones17@imperial.ac.uk*
*Department of Computing, Imperial College London, UK*

**Nick Pawlowski**                                            *npawlowski@microsoft.com*
*Microsoft Research, Cambridge, UK*
*Department of Computing, Imperial College London, UK*

**Ben Glocker**                                                 *b.glocker@imperial.ac.uk*
*Department of Computing, Imperial College London, UK*

**Reviewed on OpenReview:** *https://openreview.net/forum?id=cxp7n9q5c4*

## Abstract

Variational autoencoders (VAEs) are a popular class of deep generative models with many variants and a wide range of applications. Improvements upon the standard VAE mostly focus on the modelling of the posterior distribution over the latent space and the properties of the neural network decoder. In contrast, improving the model for the observational distribution is rarely considered and typically defaults to a pixel-wise independent categorical or normal distribution. In image synthesis, sampling from such distributions produces spatially-incoherent results with uncorrelated pixel noise, resulting in only the sample mean being somewhat useful as an output prediction. In this paper, we aim to stay true to VAE theory by improving the samples from the observational distribution. We propose SOS-VAE, an alternative model for the observation space, encoding spatial dependencies via a low-rank parameterisation. We demonstrate that this new observational distribution has the ability to capture relevant covariance between pixels, resulting in spatially-coherent samples. In contrast to pixel-wise independent distributions, our samples seem to contain semantically-meaningful variations from the mean allowing the prediction of multiple plausible outputs with a single forward pass.

## 1 Introduction

Generative modelling is one of the cornerstones of modern machine learning. One of the most used and widespread classes of generative models is the Variational Autoencoder (VAE) (Kingma & Welling, 2014; 2019). VAEs explicitly model the distribution of observations by assuming a latent variable model with low-dimensional latent space and using a simple parametric distribution in observation space. Using a neural network, VAEs decode the latent space into arbitrarily complex observational distributions.

Despite many improvements on the VAE model, one often-overlooked aspect is the choice of observational distribution. As an explicit likelihood model, the VAE assumes a distribution in observation space – using a delta distribution would not allow gradient based optimisation. Most current implementations, however, employ only simple models, such as pixel-wise independent normal distributions, which eases optimisation

but limits expressivity. Else, the likelihood term is often replaced by a reconstruction loss – which, in the case of an $L_2$ loss, implicitly assumes an independent normal distribution. Following this implicit assumption, samples are then generated by only predicting the mean, rather than sampling in observation space.

An application where this disconnect becomes apparent is image synthesis. The common choices for observational distributions are pixel-wise independent categorical or normal distributions. For pixel-wise independent distributions, regardless of other model choices, sampling from the joint distribution over pixels will result in spatially-incoherent samples due to independent pixel noise (cf. Figure 1). To address this problem, researchers use the predicted distributions to calculate the log-likelihood in the objective but then discard them in favour of the mean when generating samples or reconstructing inputs. However, this solution is akin to ignoring the issue rather than attempting to solve it.

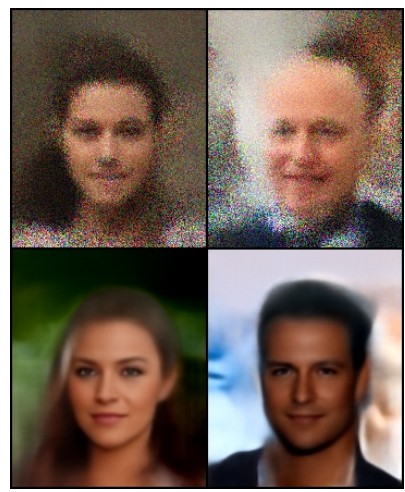

In this work, we explore what happens when we strictly follow VAE theory and sample from the predicted observational distributions. We illustrate the problem of spatial incoherence that arises from using pixel-wise independent distributions. We propose SOS-VAE, using a spatially dependent joint distribution over the observation space and compare it to the previous scenario. We further compare the samples to the mean of the predicted observational distribution, which is typically used when synthesising images. We note that, in this work, we are not focusing on absolute image quality. Instead, we aim to point to an issue often overlooked in VAE theory and application, which affects most state-of-the-art methods. Thus we

Figure 1: **Top:** Samples generated with a $\beta$-VAE exhibiting pixel-wise independent noise. **Bottom:** Samples with our SOS-VAE are realistic and spatially coherent.

analyse the relative difference between using and not using a joint pixel dependent observational distribution for a basic VAE. Yet, our findings are of broad relevance and our proposed observational distribution can be used in more advanced VAE variants, such as the NVAE (Vahdat & Kautz, 2020).

## 2 Related Work

Modern generative models can be divided into two classes, implicit likelihood models, such as Generative Adversarial Networks (GANs) (Goodfellow et al., 2014) and diffusion models (Song & Durkan, 2021), and explicit likelihood models, such as VAEs (Kingma & Welling, 2014; Rezende et al., 2014; Kingma & Welling, 2019), flow models (Dinh et al., 2017; Kingma & Dhariwal, 2018; Dinh et al., 2015) and auto-regressive models (Van Den Oord et al., 2016a;b; Salimans et al., 2017). Despite implicit likelihood models having achieved impressive results in terms of sample quality without the need for explicitly modelling the observation space (Karras et al., 2020; Brock et al., 2019), interest in explicit likelihood models have prevailed due to their appealing properties, such as ease of likelihood estimation.

One of the most popular and successful explicit likelihood models is the VAE (Kingma & Welling, 2014; Rezende et al., 2014; Kingma & Welling, 2019). Since its introduction, there have been numerous extensions. For example, Van Den Oord et al. (2017); Razavi et al. (2019) quantize the latent space to achieve better image quality; Higgins et al. (2017); Chen et al. (2017) modify the latent posterior to obtain disentangled and interpretable representations; and Vahdat & Kautz (2020); Sønderby et al. (2016) use hierarchical architectures to improve sample quality. A prominent example is the NVAE (Vahdat & Kautz, 2020) whose output distribution produces samples that are sharp and of high quality. The distribution that is being used, however, is still pixel-wise independent (discretized logistic mixture), and cannot model spatially coherent samples in the observation space. The very nature of a pixel-wise independent distribution does not allow it to model spatial dependencies between pixels. Any variation to the mean remains uncorrelated, which can be defined as noise even if it is not visually apparent. The success of the NVAE's high quality samples largely stems from the hierarchical architectural design, using spatial latent variables and other design choices that result in spatial coherence, but not the pixel-wise independent distribution, which remains to

have disadvantages that our work addresses. SOS-VAEs are complementary to the contributions in NVAE, addressing a different problem that persists in these recent approaches. Thus, our contributions such as the ability to generate multiple plausible predictions and allowing efficient interactive editing with a single forward pass stand independently to recent architectural contributions.

Like most other explicit likelihood models, the VAE requires the choice of a parametric observational distribution. This choice is often pixel-wise independent. As a result, practitioners use the distribution to calculate the likelihood but use its expected value when sampling, as the samples themselves, are noisy and of limited use in most applications. However, according to the theory and as previously pointed out by Stirn & Knowles (2020) and Detlefsen et al. (2019), a latent space sample should entail a distribution over observations and not a single point. Despite attempts to enforce spatial dependencies in the decoder architecture (Miladinovic et al., 2021), without a pixel-dependent joint likelihood, the observation samples will remain noisy. Notable exceptions are auto-regressive models and auto-regressive VAE decoders (Van Den Oord et al., 2017; Razavi et al., 2019; Gulrajani et al., 2017; Nash et al., 2021). Unlike other explicit likelihood models, auto-regressive models jointly model the observational distribution by sequentially decoding pixels while conditioning on previously decoded values. While this sampling procedure results in spatially coherent samples, it is computationally expensive and uncertainty estimation is not trivial.

In the context of non-auto-regressive VAE decoders, work focusing on modelling a joint observational distribution that accounts for pixel dependencies is limited. Monteiro et al. (2020) use a low-rank multivariate normal distribution to produce spatially consistent samples in a segmentation setting. However, they focus on discriminative models only. For generative models, a notable exception is a work by Dorta et al. (2018) which, similarly to our proposed method, employs a non-diagonal multivariate normal distribution over observation space. The key difference is the choice of parameterisation used for the covariance matrix. Dorta et al. (2018) predict the Cholesky-decomposed precision matrix, which grows quadratically with the size of the image. To address this computational constraint, the authors use a sparse decomposition. The implementation of sparse matrices incurs additional overhead and complexity required to avoid a conversion to dense matrices. Results from our head-to-head comparison show quantitative and qualitative improvements for our method compared to Dorta et al. (2018).

## 3 Methods

### 3.1 Variational Autoencoders

We start by briefly revising the theory of the standard VAE as proposed by Kingma & Welling (2014) and Rezende et al. (2014). Given a random variable $\mathbf{x}$ and its latent representation $\mathbf{z}$ whose posterior $p(\mathbf{z}|\mathbf{x})$ is intractable, the aim of the VAE is to obtain efficient inference and learning for the generative model $p(\mathbf{x}|\mathbf{z})$. Since the latent posterior is intractable, the VAE uses variational inference to approximate the posterior $p(\mathbf{z}|\mathbf{x})$ using a variational distribution $q(\mathbf{z}|\mathbf{x})$. The approximate latent posterior and the generative model are parameterised by neural networks. For a dataset $\mathbf{X} = \{\mathbf{x}^{(i)}\}_{i=1}^{N}$, the VAE objective for a single data-point is given by maximising the evidence lower bound with respect to the networks' parameters:

$$\mathcal{L}(\boldsymbol{\theta}, \boldsymbol{\phi}; \mathbf{x}^{(i)}) = -D_{KL}\left[q_{\boldsymbol{\phi}}(\mathbf{z}|\mathbf{x}^{(i)})||p(\mathbf{z})\right] + \mathbb{E}_{q_{\boldsymbol{\phi}}(\mathbf{z}|\mathbf{x}^{(i)})}\left[\log p_{\boldsymbol{\theta}}(\mathbf{x}^{(i)}|\mathbf{z})\right], \tag{1}$$

where $p(\mathbf{z})$ is the latent prior, $q_{\boldsymbol{\phi}}(\mathbf{z}|\mathbf{x}^{(i)})$ is the latent variational posterior parameterised by a neural network with parameters $\boldsymbol{\phi}$, and $p_{\boldsymbol{\theta}}(\mathbf{x}^{(i)}|\mathbf{z})$ is the posterior likelihood parameterised by parameters $\boldsymbol{\theta}$. The latent prior and variational latent posterior are chosen to be normally distributed such that the KL divergence can be computed in closed form. The likelihood cannot be computed in closed form since the derivative of the lower bound w.r.t $\boldsymbol{\phi}$ is problematic due to the stochastic expectation operator. Consequently, Monte-Carlo estimation and the re-parameterisation trick are used to yield the Stochastic Gradient Variational Bayes (SGVB) estimator (Kingma & Welling, 2014).

### 3.2 The Problem with Images

The likelihood term in the ELBO (equation 1) requires the choice of a parametric observational distribution to calculate the likelihood that the data comes from the predicted distribution: $p_{\boldsymbol{\theta}}(\mathbf{x}|\mathbf{z})$. It follows that each

latent sample entails a distribution over observations and that the predicted distribution should be as close as possible to the observed distribution for its samples to look like real observations. The standard choice of the observational distribution for image data is pixel-wise independent Bernoulli or Gaussian distributions since these make the likelihood calculation straightforward. However, pixel-wise independent distributions ignore the strong spatial structure in the data, therefore, are poor choices for observational distributions in images. This is evidenced by practitioners ignoring sampling from the observational distribution in favour of its expected value to avoid pixel-wise independent noise from corrupting samples, as shown in figure 1. This workaround technique has become the de facto standard for generating images with VAEs, despite theory describing the modelling of an observed distribution, not just its expected value. Models that only use the expected value are deviating from theory to cover for an inadequate choice of observational distribution. By incorporating spatial dependencies in the predicted distribution, we aim to overcome this limitation and generate more realistic samples under the observed distribution, without relying only on the expected value.

### 3.3 Structured Observation Space VAE

We propose replacing the joint pixel-wise independent distributions used in the observational posterior with a distribution that explicitly models the dependencies between pixels. Specifically, following recent developments in discriminative models for segmentation (Monteiro et al., 2020), we use a multivariate normal distribution with a fully populated covariance matrix $p_{\boldsymbol{\theta}}(\mathbf{x}|\mathbf{z})=\mathcal{N}(\boldsymbol{\mu}, \boldsymbol{\Sigma})$, which we compute efficiently using a low-rank parameterisation. This small modification can be applied to most existing VAE architectures.

Given an image $\mathbf{x}$ or sample $\mathbf{y}$ with width $W$, height $H$, and $C$ channels, we consider its flat representation with size $S = W \cdot H \cdot C$ and use a three-headed VAE decoder which outputs:

- The mean of the multivariate normal distribution $\boldsymbol{\mu}$ with size $S$;

- The elements $\mathbf{d}$ of a positive diagonal matrix $\mathbf{D}$ with $S$ diagonal elements;

- The covariance factor matrix $\mathbf{P}$ with size $S \times R$ where $R$ is the rank hyperparameter chosen for the parameterisation.

The low-rank covariance matrix can be computed as $\boldsymbol{\Sigma} = \mathbf{P}\mathbf{P}^T + \mathbf{D}$ which has $S + (S \times R)$ degrees of freedom as opposed to $S^2$ for a full-rank covariance matrix. The rank hyperparameter is a trade-off between expressivity and computational cost. As a hyperparameter, the only unbiased way of selecting is through ablation. In practice, we found it sensible to select it based on the available computation resources. To calculate the likelihood of a multivariate normal distribution, we must invert the covariance matrix and compute its determinant. For a full-rank covariance matrix, the computational cost and memory footprint make this infeasible for all except very small images since these costs scale quadratically with image size. However, with the low-rank parametrisation, the full matrix never needs to be used directly. The inverse and determinant of the covariance matrix can be efficiently computed using the Woodbury matrix identity Woodbury (1950) and the matrix determinant lemma, respectively. Since both the covariance factor $\mathbf{P}$ and the diagonal elements $\mathbf{d}$ scale linearly with the size of the image, the computational costs of computing the log-likelihood scales linearly as well, in contrast to a full-rank covariance matrix where costs increase quadratically.

The proposed modification on the likelihood distribution does not affect the theory of the VAE. Thus the SGVB estimator (equation 1) is applicable without modification. However, we found optimising the SGVB estimator with a free non-diagonal covariance to be unstable due to exploding variance. The instability comes from there being two routes that minimise the likelihood. Either to find an appropriate mean and variance around it or to keep increasing the variance (uncertainty about the mean). The second route is undesirable and results in implausible samples (with overly bright colours and high contrast; cf. Appendix A). This problem is not unique to our implementation; the stability of variance networks has been discussed before (Stirn & Knowles, 2020; Detlefsen et al., 2019). To address this issue, we introduce an entropy penalty on the observational distribution. Constraining the entropy constrains the variance indirectly, thus giving preference to low-variance solutions. Therefore, we compute the entropy of the normal distribution in closed

form and add it to the objective function. The new objective, now also including a $\beta$ penalty (Higgins et al., 2017) is given in by:

$$\tilde{\mathcal{L}}(\boldsymbol{\theta}, \boldsymbol{\phi}; \mathbf{x}^{(i)}) = -\beta D_{KL}\left[(q_{\boldsymbol{\phi}}(\mathbf{z}|\mathbf{x}^{(i)})||p_{\boldsymbol{\theta}}(\mathbf{z}))\right] + \mathbb{E}_{q_{\boldsymbol{\phi}}(\mathbf{z}|\mathbf{x}^{(i)})}\left[\log p_{\boldsymbol{\theta}}(\mathbf{x}^{(i)}|\mathbf{z})\right] - H(\mathbf{x}^{(i)}|\mathbf{z}), \qquad (2)$$

where $H(\mathbf{x}^{(i)}|\mathbf{z})$ is the entropy of the predicted observational distribution. In addition to the entropy constraint, we used the following optimisation tricks to further incentivise the low variance solution: we freeze the variance heads of the decoder and train the decoder with only the mean head for a few epochs Monteiro et al. (2020); we fix the covariance diagonal to a small positive scalar in the low-rank parametrisation $\mathbf{D} = \epsilon \mathbf{I}$[1].

Multiple-constraint optimisation can lead to bad convergence properties, as a result, we opted to employ soft-constraints for the entropy and KL divergence using the modified differential method of multipliers (Platt & Barr, 1988), for its agreeable convergence and stability properties, which results in the following Lagrangian formulation:

$$
\begin{aligned}
\tilde{\mathcal{L}}(\boldsymbol{\theta}, \boldsymbol{\phi}; \mathbf{x}^{(i)}) = &+\mathbb{E}_{q_{\boldsymbol{\phi}}(\mathbf{z}|\mathbf{x}^{(i)})}\left[\log p_{\boldsymbol{\theta}}(\mathbf{x}^{(i)}|\mathbf{z})\right] \\
&- \beta\left[D_{KL}\left[(q_{\boldsymbol{\phi}}(\mathbf{z}|\mathbf{x}^{(i)})||p_{\boldsymbol{\theta}}(\mathbf{z}))\right] - \xi_{KL}\right] \\
&- \lambda_H\left[H(\mathbf{x}^{(i)}|\mathbf{z}) - \xi_H\right]
\end{aligned}
\qquad (3)
$$

where $\beta$ and $\xi_{KL}$ are the Lagrangian multiplier and the slack variable, respectively, for the $\beta$-VAE constraint and $\lambda_H$ and $\xi_H$ are the Lagrangian multiplier and the slack variable, respectively, for the entropy constraint. The slack variables are tunable parameters of the model. Since they have semantic values, they can be chosen empirically rather than through blind trial and error. The slack variables for the entropy and KL constraints can be selected by observing the respective values from an unconstrained model. These observations give the orders of magnitude for the slack variables before further tuning. For our SOS-VAE, training remains stable for slack variables in the correct order of magnitude, however we found the entropy constraint to be detrimental to the training of a $\beta$-VAE, which can be explained by the decreased expressivity of the $\beta$-VAE compared to our SOS-VAE.

The prior work of Dorta et al. (2018) uses a non-diagonal covariance matrix for the distribution over the observation space, however our choice of parameterisation differs, as discussed in section 2. Furthermore, our choice to use the modified differential method of multipliers differs and our introduction of an entropy constraint is novel, as best we know, for this use case.

## 4 Experiments and results

### 4.1 $\beta$-VAE vs. Structured Observation Space VAE

We start by comparing a $\beta$-VAE, with a pixel-wise independent normal observational distribution, to the proposed SOS-VAE method, with a low-rank multivariate normal observational distribution. We choose a $\beta$-VAE (Higgins et al., 2017) baseline rather than a standard VAE so that it can be trained using the same Lagrangian method with $\beta$ being a Lagrangian multiplier. This allows for fairer testing since the comparison is between models that differ only by our contributions. This is the only modification to the baseline over a standard VAE. We perform the comparison in two datasets: the CELEBA dataset (Liu et al., 2015) and the UK Biobank (UKBB) Brain Imaging dataset (Miller et al., 2016). For all models, we use a latent space of dimension 128. For the low-rank model, we use a rank of 25. For the CELEBA dataset we use a target KL loss , $\xi_{KL}$, of 45 for both models and $\xi_H = -504750$ for our model. For the UKBB dataset we use a target KL loss, $\xi_{KL}$, of 15 for both models and $\xi_H = -198906$ for our model. Figures 2a & 2b and 2c & 2d show the qualitative results for the comparison.

In Figures 2a and 2c, we see that the samples of the $\beta$-VAE exhibit uncorrelated pixel noise around the mean, resulting from the pixel-wise independent joint observational distribution. In contrast, in Figures 2b and 2d,

---

[1]We found $10^{-5}$ to yield good results.

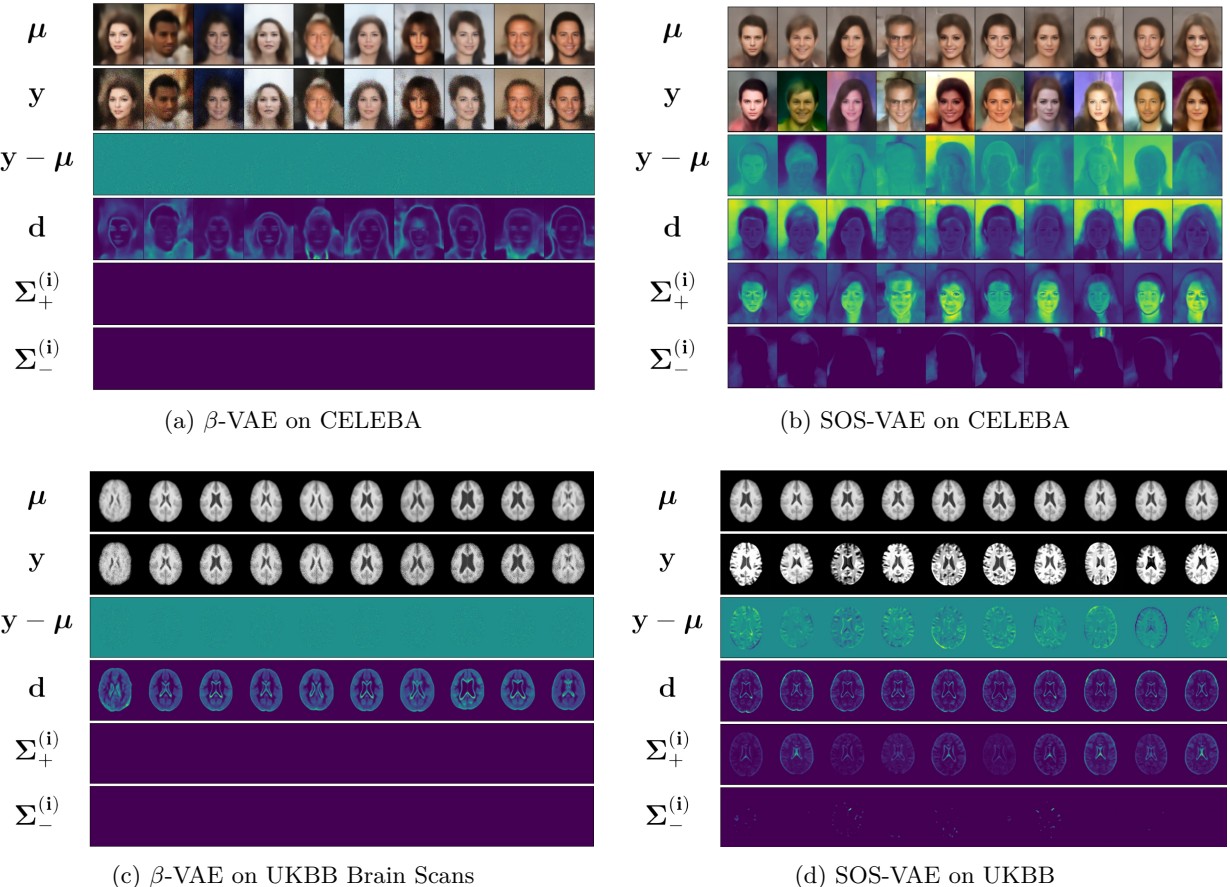

Figure 2: Qualitative results comparing a $\beta$-VAE (2a) and SOS-VAE (2b) on the CELEBA dataset. Same comparison on the UKBB dataset (2c & 2d). The rows from top to bottom: the mean of the observational distribution, a sample from the observational distribution, the difference between the mean and the given sample, the pixel-wise independent variance per pixel, a slice of the covariance matrix: positive covariance and negative covariance to the central pixel.

Dorta et al.                                    Ours

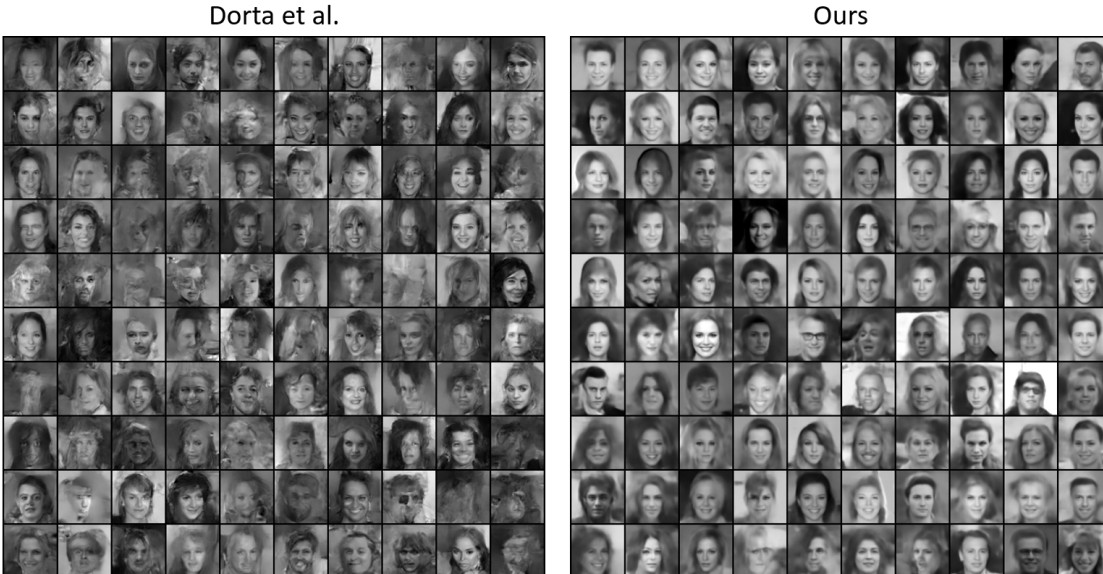

Figure 3: A comparison of 100 64x64 grayscale samples from a Structured Uncertainty Prediction Network VAE (Dorta et al., 2018) (left) and from SOS-VAE (right).

we see that the samples produced by our SOS-VAE contain semantically-meaningful variations around the mean and are spatially coherent, as illustrated in the difference (row 3) between the mean (row 1) and the sample (row 2). Looking at the variance of the two methods (row 4), we see a significant difference in the regions where each model is uncertain, highlighting the difference in behaviour between the two predicted distributions. The predicted covariance (rows 5 and 6) for the low-rank model contains a structure that pertains to the image content. These figure rows represent positive and negative covariance to the central pixel indicating global covariance can is modelled. This structure results in spatially coherent samples as opposed to the noisy samples of the $\beta$-VAE, which are a consequence of the diagonal covariance. Interestingly, we observe more variation in the means of the $\beta$-VAE, suggesting that as more variation can be modelled in the observation space, less needs to be modelled in the latent space.

Quantitative evaluation of generative modelling is an inherently difficult task due to its subjective nature. While measuring the log-likelihood is the obvious choice, it is often not indicative of sample quality (Theis et al., 2016; Borji, 2019). The Fréchet Inception Distance (FID) (Heusel et al., 2017) is the current standard choice of metric due to its consistency with human perception (Borji, 2019). We note this metric is not without criticism (Borji, 2019; Razavi et al., 2019). We use it to evaluate our generative models (Seitzer, 2020) and report the results in Table 1. As in (Heusel et al., 2017), we generate $50,000$ samples from each of our models and propagate them, in turn, through the Inception-v3 model to sufficiently represent the set of synthesizable samples. This reduces sample bias to produce reliable results.. The results do not represent the absolute performance of SOS-VAE, but rather the relative difference when compared to a $\beta$-VAE with a pixel-wise independent observational distribution while everything else is constant. We emphasise, the proposed method is compatible with generative models from other works. We observe that, for both datasets, the proposed SOS-VAE achieves a lower FID score than the $\beta$-VAE. Notably, the samples from the model with a low-rank multivariate normal observational distribution outperform the means of the $\beta$-VAE, indicating that sampling from the observational distribution, as theory entails, does not reduce image quality.

Table 1 also reports the results for the cropped and aligned CELEBA dataset, resized to 64x64 and converted to grayscale[2]. This enables the comparison to Dorta et al. (2018)'s Structured Uncertainty Prediction Network trained with equivalent parameters. Note that the image size and number of colour channels is constrained by the code of Dorta et al. (2018). We used our own implementation of Dorta et al. (2018)'s model predominantly using their original code, the same dataset and learning rate, trained for the same

---

[2]For this dataset, our method uses $\xi_H = -17630$.

Table 1: FID metric results for a $\beta$-VAE with a pixel-wise independent observational distribution, a Structured Uncertainty Prediction Network (Dorta et al., 2018) and our SOS-VAE. Lower FID scores represent better performance.

| Method | Dataset | FID $\downarrow$ |
|---|---|---|
| $\beta$-VAE (MEANS) | | 121.65 |
| $\beta$-VAE (SAMPLES) | CELEBA | 196.40 |
| SOS-VAE (MEANS) | | 132.93 |
| SOS-VAE (SAMPLES) | | **104.62** |
| $\beta$-VAE (MEANS) | | 211.24 |
| $\beta$-VAE (SAMPLES) | UKBB | 332.89 |
| SOS-VAE (MEANS) | | 141.87 |
| SOS-VAE (SAMPLES) | | **79.712** |
| $\beta$-VAE (MEANS) | | 90.97 |
| $\beta$-VAE (SAMPLES) | | 257.11 |
| DORTA ET AL. (MEANS) | 64x64 | 73.44 |
| DORTA ET AL. (SAMPLES) | GRAYSCALE | 77.72 |
| SOS-VAE (MEANS) | CELEBA | **70.43** |
| SOS-VAE (SAMPLES) | | 74.66 |

number of epochs. Using the FID metric, we find that our SOS-VAE outperforms Dorta et al. (2018) in both means and samples respectively. Qualitatively comparing samples from Dorta et al. (2018) and SOS-VAE (Figure 3), both exhibit spatially-correlated variations from the mean. However, in the samples generated with the method by Dorta et al. (2018), the variations exhibit high-frequency distortions. In contrast, the samples obtained with our SOS-VAE show semantically-meaningful, spatially-correlated variations globally across the generated image.

## 4.2 Interpolation in the Observation Space

To explore the expressiveness of the representations captured in the observation space, we visualise a continuous range of samples from the proposed method. We perform spherical linear interpolation over the observation space to capture a range of plausible images between two initial samples, $\mathbf{y}_a$ and $\mathbf{y}_b$. This is shown in equation 4, where $\boldsymbol{\omega}_p \in \mathbb{R}^R$ and $\boldsymbol{\omega}_d \in \mathbb{R}^{(S \times C)}$ are both auxiliary noise variables, slerp is the spherical interpolation function (see appendix D) and $t \in [0,1]$ is the interpolation factor.

$$\mathbf{y}_t = \boldsymbol{\mu} + \mathbf{P}\boldsymbol{\omega}_{p_t} + \sqrt{\epsilon}\,\boldsymbol{\omega}_d \tag{4}$$
$$\text{where } \boldsymbol{\omega}_{p_t} = \text{slerp}(\boldsymbol{\omega}_{p_a}, \boldsymbol{\omega}_{p_b}, t)$$

It is important to note that we only interpolate over $\boldsymbol{\omega}_p$ as it restricts the dimensionality of the hyper-sphere to size $R$; the $\sqrt{\epsilon}\,\boldsymbol{\omega}_d$ term only adds a small amount of uncorrelated noise, so setting it as a constant has negligible effect. Figure 4 shows this technique to visualise the variation contained in the observation space, demonstrating that the distribution captures semantically-relevant features, such as hair colour, skin tone and background colour for the CELEBA dataset. Interpolation between differences in such features would typically entail interpolating latent variables and a forward pass through the decoder for each interval, which is not required here.

## 4.3 Interactive Sampling from the Observation Space

We have demonstrated that a low-rank multivariate normal observational distribution can model a range of features. However, it would be useful if we could synthesise images with semantically-meaningful human input. A step in this direction entails fixing the auxiliary noise variables associated with each sample and scaling the principal components of our covariance factor, $\mathbf{P}$. This is demonstrated in equation 5: using

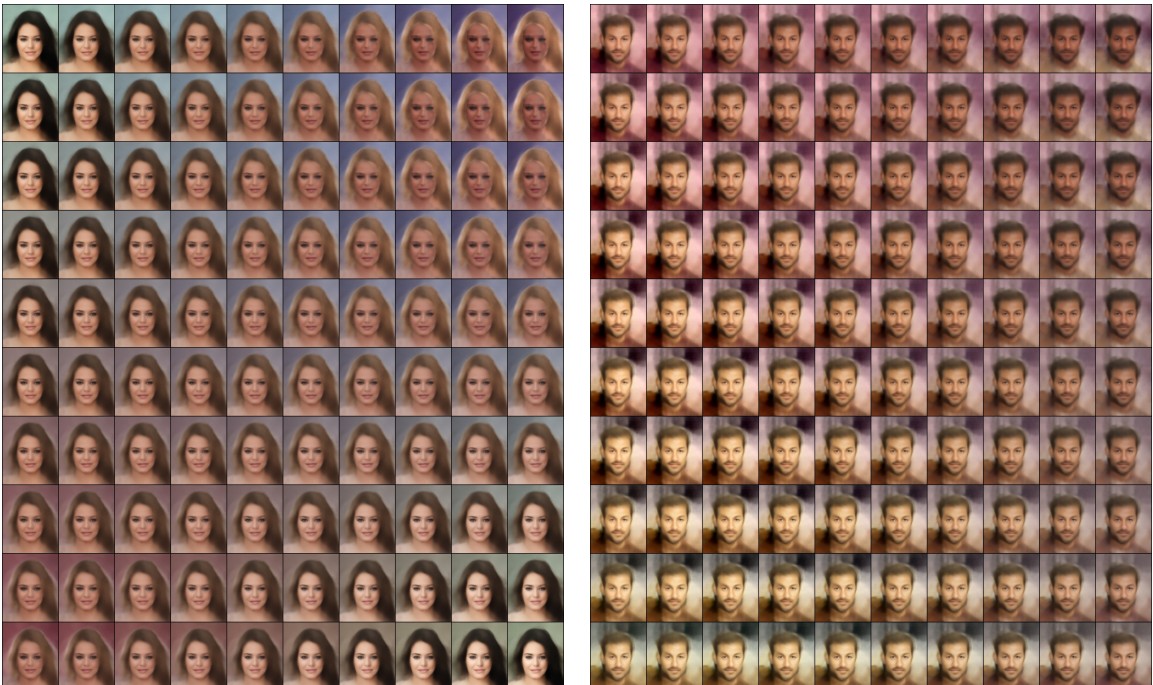

Figure 4: Spherical linear interpolation between auxiliary noise variables $\boldsymbol{\omega}_p$. The four corners are random samples from a predicted distribution with all intermediate steps as interpolations between them. The two images represent interpolations in the observation space for two observational distributions predicted from different latent codes.

the singular value decomposition (SVD) of $\mathbf{P}$ and introducing a diagonal matrix of scaling coefficients, $\mathbf{A} \in \mathbb{R}^{(R \times R)}$.

$$\mathbf{P} = \mathbf{U}(\mathbf{SA})\mathbf{V}^T \tag{5}$$

Adjusting the scaling coefficients in $\mathbf{A}$ allows us to tune spatially correlated features in the sample image. The use of SVD often makes the effect of each coefficient separable and semantically-relevant to the image domain. Images generated through this method for CELEBA are shown in Figure 5, where each row demonstrates the effect of scaling a different principal component. This figure shows the effect on the first ten principal components. The effect on all components and additional results on the UKBB data are given in Appendix B. Since this manipulation is using only the observational distribution, manipulation of these samples can be achieved without performing additional forward passes on the model.

## 4.4 Interactive Editing of Predictions

One of the benefits of modelling spatial correlations in the observational distribution is that this information can be leveraged to interactively edit predictions. This involves manually editing part of the prediction and calculating the conditional distribution of the remaining pixels; for an arbitrary multivariate normal distribution, this is expressed in equations 6 and 7, where the edited pixels are modelled by $\mathbf{y}_2$ and the remaining pixels are modelled by $\mathbf{y}_1$. We use the mean of the conditional distribution ($\tilde{\boldsymbol{\mu}}$) as the corrected image. The updated covariance matrix ($\tilde{\boldsymbol{\Sigma}}$) is not needed, so we avoid evaluating it to reduce the computational cost of this process.

$$\mathbf{y} = \begin{bmatrix} \mathbf{y}_1 \\ \mathbf{y}_2 \end{bmatrix}, \quad \boldsymbol{\mu} = \begin{bmatrix} \boldsymbol{\mu}_1 \\ \boldsymbol{\mu}_2 \end{bmatrix}, \quad \boldsymbol{\Sigma} = \begin{bmatrix} \boldsymbol{\Sigma}_{11} & \boldsymbol{\Sigma}_{12} \\ \boldsymbol{\Sigma}_{21} & \boldsymbol{\Sigma}_{22} \end{bmatrix} \tag{6}$$

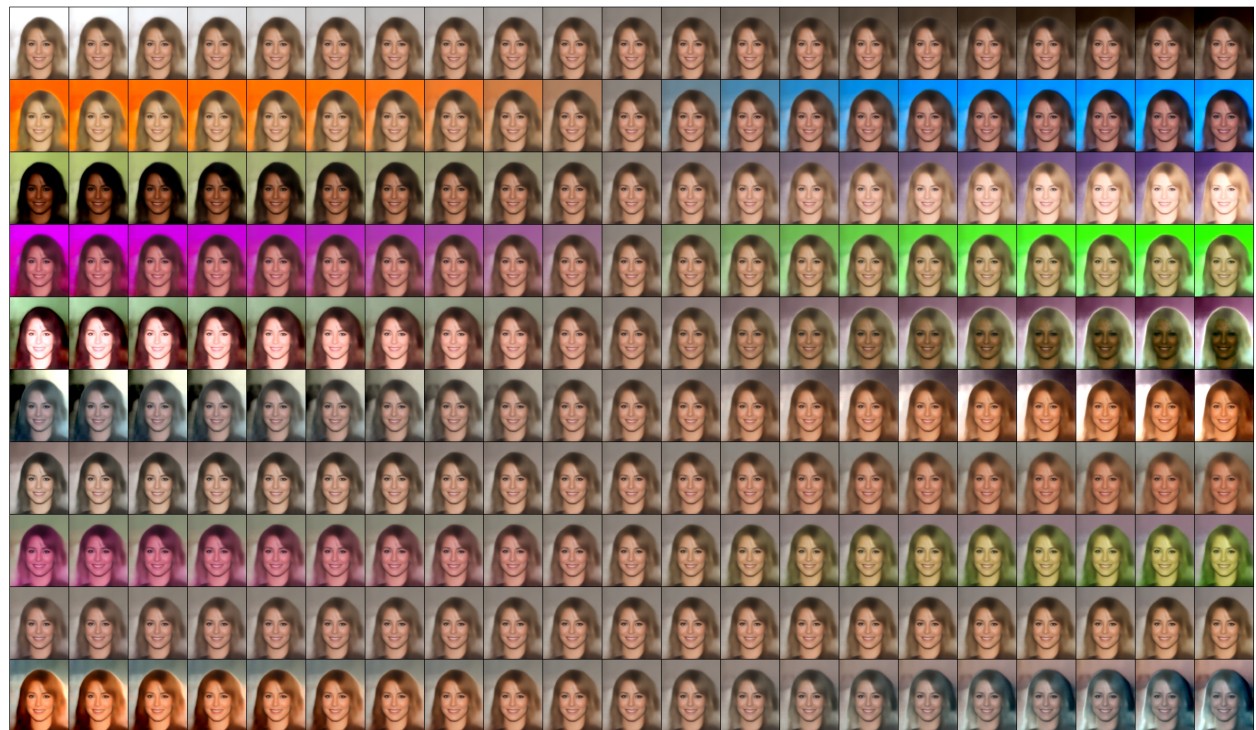

Figure 5: The effect of scaling each of the ten most principal components (each row), from top to bottom for a fixed auxiliary noise variable. The scale factor for each component ranges from $-5$ to $+5$ with intervals of 0.5.

Interactive editing is demonstrated in Figure 6 (with additional examples in Appendix C), where a prediction from our model, trained on the CELEBA dataset, is sequentially edited to alter the hair colour and skin tone. This demonstrates the power of the method: we can manually edit a small number of pixels and automatically update the remainder of the image coherently with the manual edit.

$$p(\mathbf{y}_1|\mathbf{y}_2 = \mathbf{b}) = \mathcal{N}(\tilde{\boldsymbol{\mu}}, \tilde{\boldsymbol{\Sigma}}) \tag{7}$$
$$\text{where } \tilde{\boldsymbol{\mu}} = \boldsymbol{\mu}_1 + \boldsymbol{\Sigma}_{12}\boldsymbol{\Sigma}_{22}^{-1}(\mathbf{b} - \boldsymbol{\mu}_2)$$
$$\text{and } \tilde{\boldsymbol{\Sigma}} = \boldsymbol{\Sigma}_{11} - \boldsymbol{\Sigma}_{12}\boldsymbol{\Sigma}_{22}^{-1}\boldsymbol{\Sigma}_{21}$$

### 4.5 Observational Distribution without Deep Learning

Typical VAEs rely on their deep learning components to model the features for their output. In contrast, since we use a low-rank parameterisation of a full covariance matrix, our observational distribution can model spatially-correlated features on its own. As a result, the ability to model these features is not solely left to the deep learning components of the VAE; it is shared with the linear transformations that compose the low-rank multivariate normal distribution.

To understand the expressiveness of our observation space model, we carried out an experiment in which the VAE architecture is replaced with the parameters for the low-rank multivariate normal distribution with no deep learning layers or latent space representation involved. We then train this simple model as described in section 3.3. This allows us to examine the capability of the model to capture the observational distribution over the dataset, without deep learning.

This modelling is reminiscent of principal component analysis (PCA), particularly given that we use a low-rank parameterisation, akin to the dimensionality reduction of PCA. Figure 7 compares samples from the

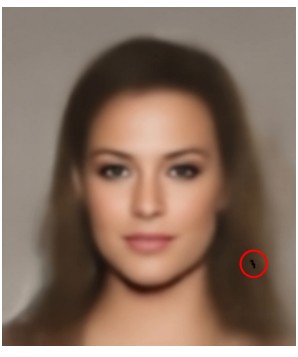 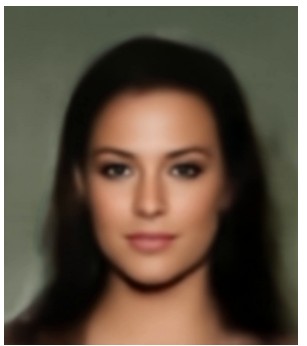 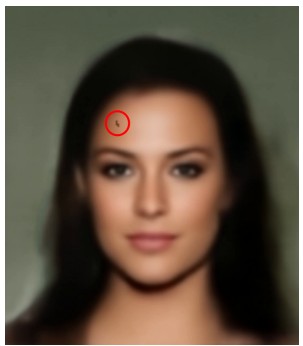 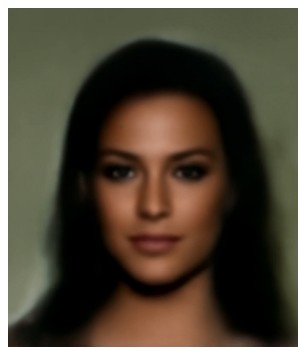

Figure 6: Sequentially editing hair colour and skin tone interactively. From left to right: a predicted image with a small coloured edit made to the hair, the image after the conditional distribution has been calculated, the image with a further edit to the skin tone, the image after the conditional distribution has been recalculated. N.B: The red circles highlighting the manual edits are for illustration purposes only and serve no computational purpose.

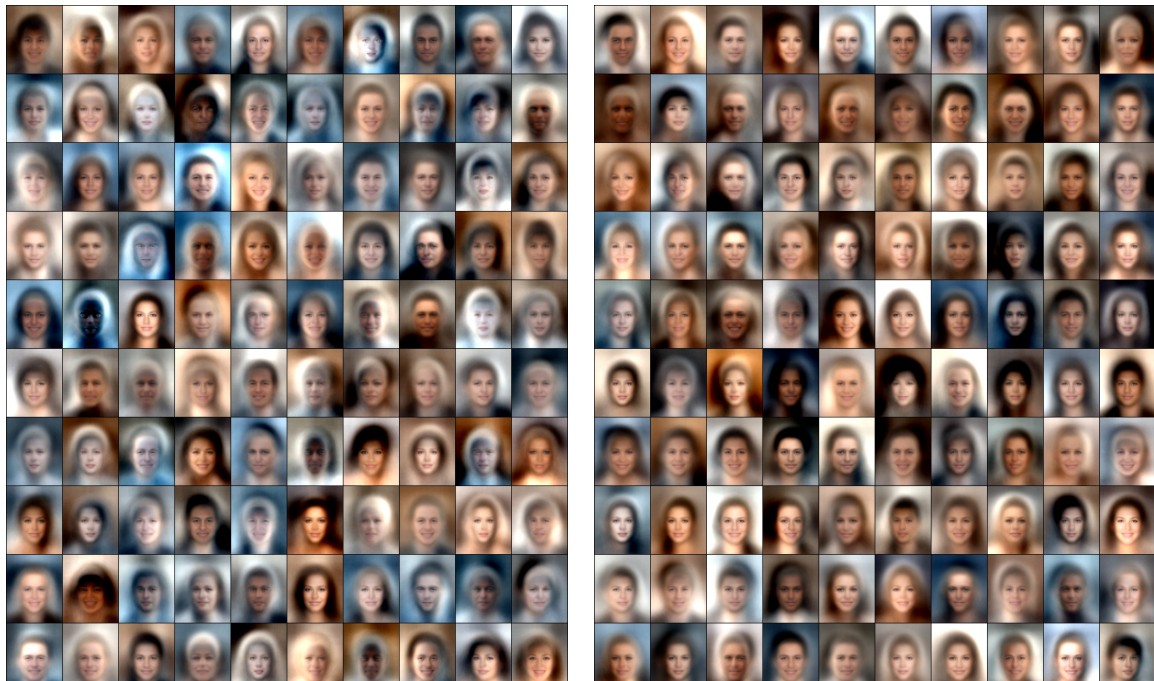

Figure 7: A qualitative comparison between 100 samples from the learnt observation space at rank = 25 with no deep learning components (left) and 100 samples from a linear PCA model with 25 features (right). In both cases, the data used for fitting is the same random subset of 10000 images from the CELEBA dataset.

distribution alongside samples from a linear PCA model of equivalent feature reduction after fitting to the CELEBA dataset (Liu et al., 2015).

There is little perceivable difference between the samples of the two methods, so we conclude that our low-rank multivariate normal distribution is comparably expressive to PCA for feature reduction. Furthermore, this experiment confirms the ability to learn the parameters of our distribution through backpropagation.

## 5 Discussion

This paper highlights an often-overlooked aspect of the VAE architecture - the observational distribution. We have confirmed that pixel-wise independent observational distributions produce samples with uncorrelated pixel noise. We have introduced SOS-VAE, a low-rank multivariate normal distribution as a choice for the observational distribution of a VAE, able to model covariance between pixels and produce multiple spatially-coherent samples with a single forward pass of the decoder. Our method introduces stability issues that are otherwise not present, but that we are able to resolve with an entropy constraint. Our results indicate that our choice of observational distribution is beneficial when compared to a pixel-wise independent distribution, as well as allowing sampling to be the primary method of image synthesis without reduction in quality, as theory would entail. Our method performs favourably compared to previous work (Dorta et al., 2018), and is compatible with many VAE architectures, including within state-of-the-art models. We find that a low-rank multivariate observational distribution can be interpolated within, and allows for semantic, interactive manipulation of samples with a single decoder forward pass from a single latent variable.

In our current experimentation we use a standard VAE architecture for ease of implementation and lower computational requirements. It would be interesting to incorporate our approach into recent models such as NVAE and other deep VAE variants (Child, 2020). However, training these models is extremely resource demanding. For the original NVAE implementation[3], the authors state that "eight 16-GB V100 GPUs are used for training NVAE on CelebA 64" and "training takes about 92 hours". In contrast, the SOS-VAE model used in our experiments is trained on a single 16-GB T4 GPU in 72 hours, which is a substantial difference of practical importance. We had considered using a pre-trained deep VAE, training only the last few layers, with and without the proposed low-rank likelihood distribution. However, we found that these models produce unsatisfactory results on lower image resolutions, and training on higher resolutions remains prohibitive with limited computational resources.

Introducing an expressive observational distribution that is able to model features on its own, as we have, promotes a discussion comparing the features modelled in the observation space to those modelled in the latent space. In section 4.5, we observe our observational distribution's ability to model features on its own and in section 4.1 we observe a decrease in variation of the predicted means for our model compared to a $\beta$-VAE. Assuming a dataset containing finite uncertainty, we deduce that the modelling of this uncertainty is split between the latent space and the observation space. Understanding where this split lies and what influences this is an open question left for future work.

**Ethical Considerations**

Generative modelling is subject to dataset-inherited bias and our contributions are also susceptible. Whilst methods, such as our own, allow us to explore the biases that exist within a dataset, which in some cases makes for a desirable tool, typical usage may expose an undesired bias, particularly after training on the CELEBA dataset, which has a larger societal impact. We acknowledge that these biases, such as lack of diversity, are present but state these as artefacts of the chosen dataset and not of our own design, opinions or beliefs. There is a clear need for more diverse and representative datasets that would allow a more complete picture of the abilities and limitations of generative models to be obtained. The current reliance on human face datasets for generative modelling is problematic. While some alternatives such as the waterbird dataset (Sagawa et al., 2019) exist, these still lack the fidelity for visual interpretation of generative modelling results. We would hope that better, alternative datasets would be available for future work.

Our method allows for the synthesis of multiple plausible samples as well as manipulation of samples with a single forward pass of the model, where other methods require multiple forward passes. This reduction in computational burden could provide access to machine learning models for those with low-powered devices as well as reducing energy consumption and computational costs.

---

[3]https://github.com/NVlabs/NVAE

## Reproducibility

All code is made publicly available on https://github.com/biomedia-mira/sos-vae together with clear instructions how to fully reproduce our results on CELEBA to facilitate future work and ease of comparisons.

## Acknowledgements

This project has received funding from the European Research Council (ERC) under the European Union's Horizon 2020 research and innovation programme (Grant Agreement No. 757173, Project MIRA).

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

## A  Out-of-the-box samples without stabilising

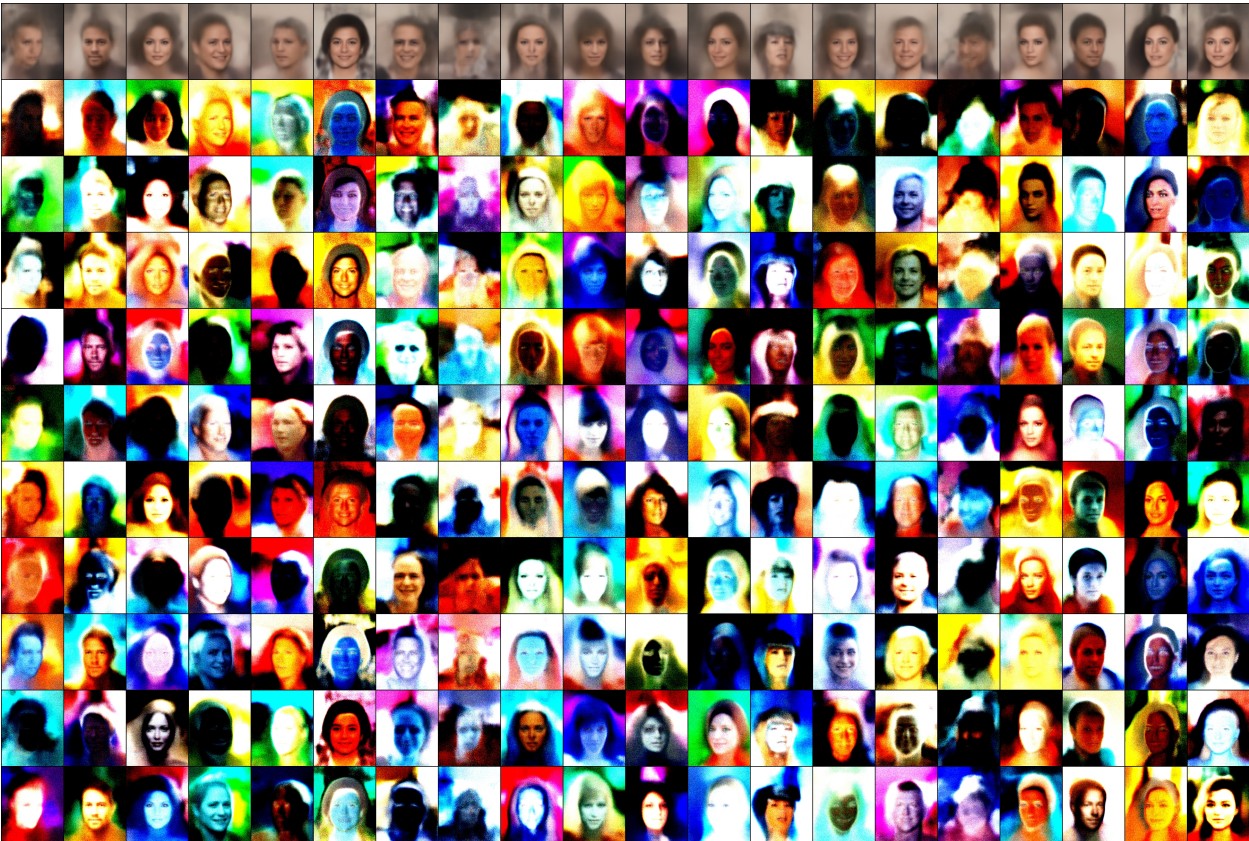

Figure 8: The results after training with a pre-training phase and weight initialisation, but without the fixed component, $\mathbf{D} = \epsilon \mathbf{I}$, or the entropy constraint. The first row represents the predicted means and all other rows represent samples from the predicted distribution outputted from the probabilistic decoder. Each column is a new sample from the latent prior decoded to predict distributions over the observation space. Model trained for 100 epochs on a random subset of 10000 images from the CELEBA dataset. Latent dimensionality: $l = 128$, rank: $R = 25$, target KL loss: $\xi_{KL} = 45$. The resulting average variance for each pixel is $\approx 11.76$, in comparison to $\approx 0.0322$ for the equivalently trained model with the entropy constraint, $\xi_H = -504750$.

## B Scaling of all individual PCA components for CELEBA and UKBB

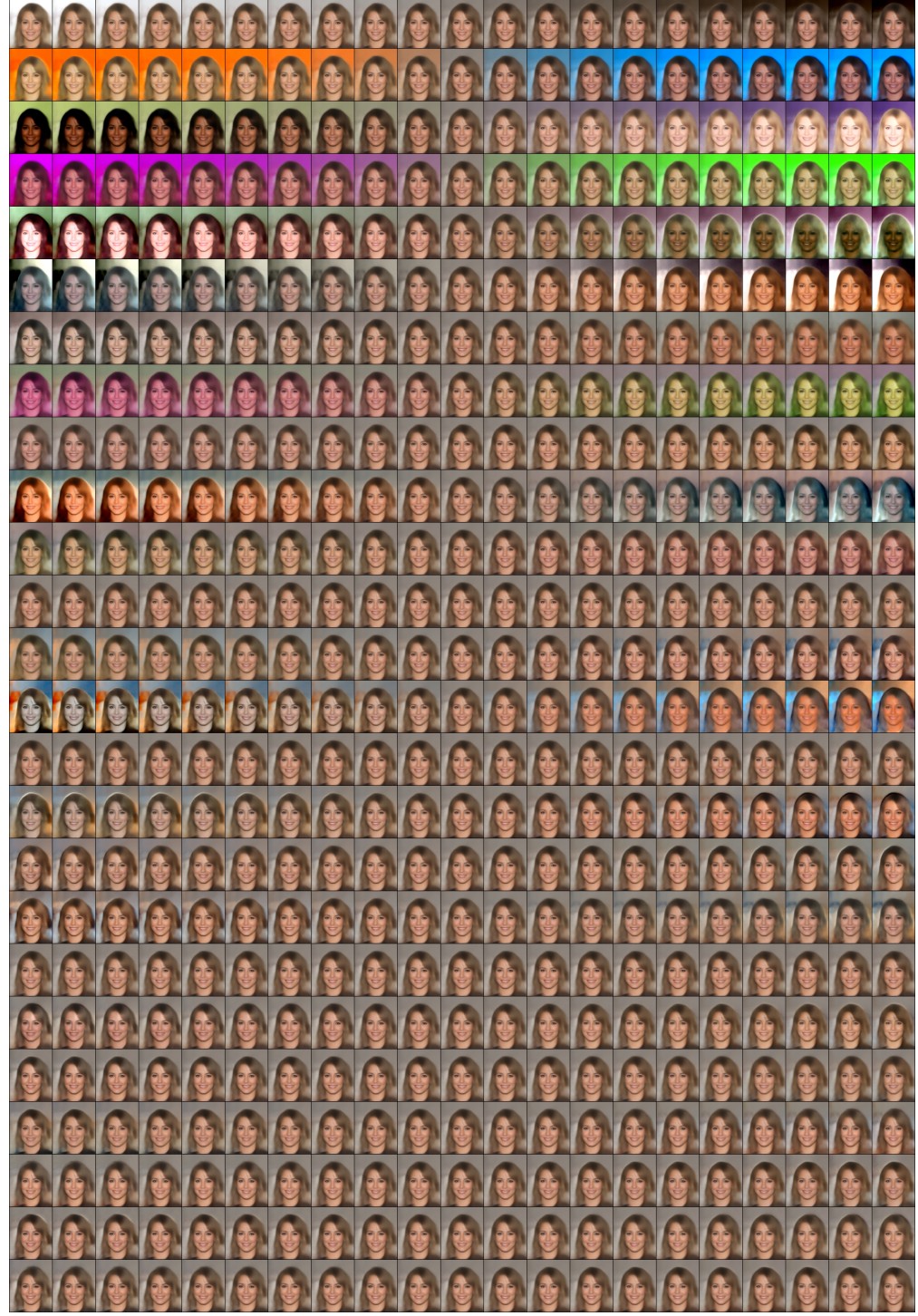

Figure 9: The effect of scaling each of the principal components (each row), from top to bottom for a fixed auxiliary noise variable and a rank 25 parameterisation. The scale factor for each component ranges from $-5$ to $+5$ with intervals of 0.5. Observational distribution predicted by our VAE after training on the CELEBA dataset.

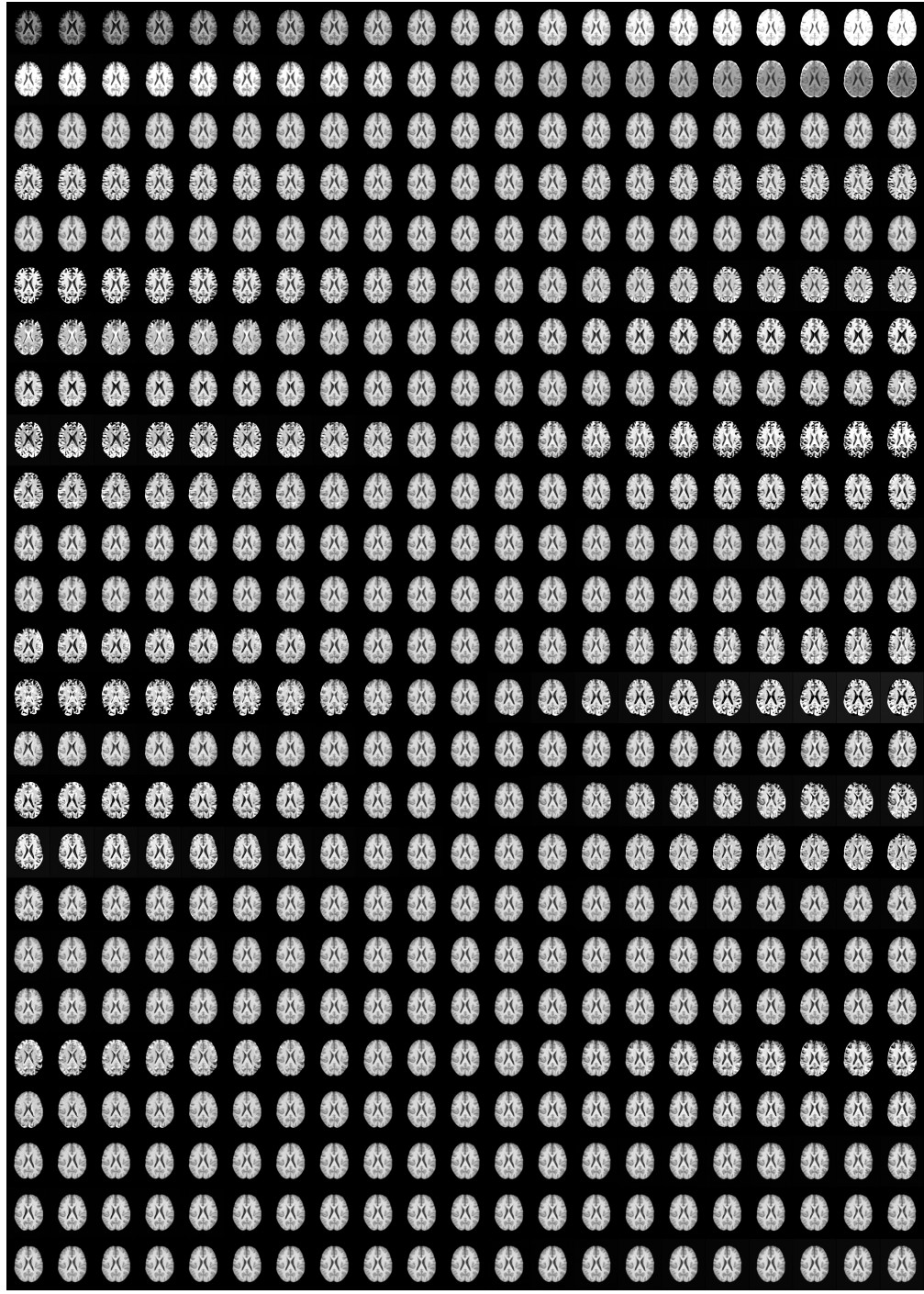

Figure 10: The effect of scaling each of the principal components (each row), from top to bottom for a fixed auxiliary noise variable and a rank 25 parameterisation. The scale factor for each component ranges from $-5$ to $+5$ with intervals of $0.5$. Observational distribution predicted by our VAE after training on the UKBB dataset.

## C   Additional Examples for Interactive Editing

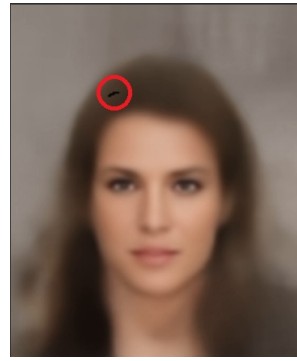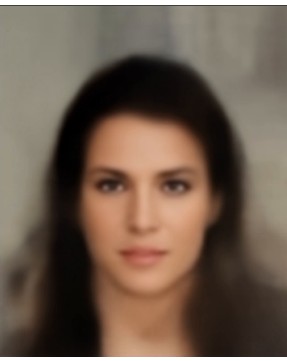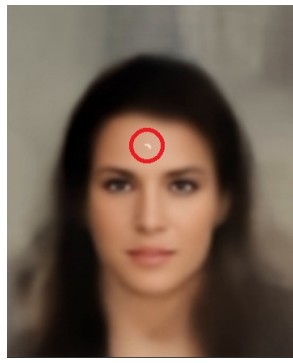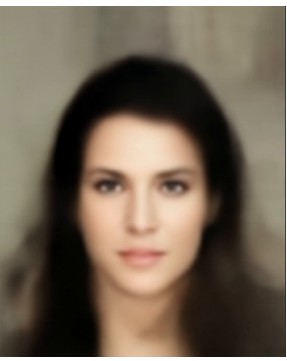

Figure 11: Sequential interactive editing. From left to right: predicted image with a small coloured edit made to the hair, the image after the conditional distribution has been calculated, the image with a further edit to the skin tone, the image after the conditional distribution has been recalculated. N.B: The red circles highlighting the manual edits are for illustration purposes only and serve no computational purpose.

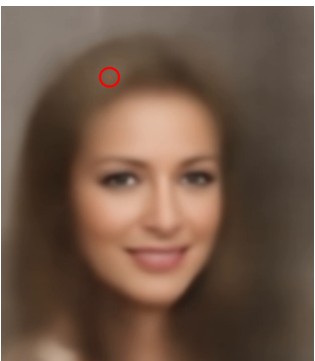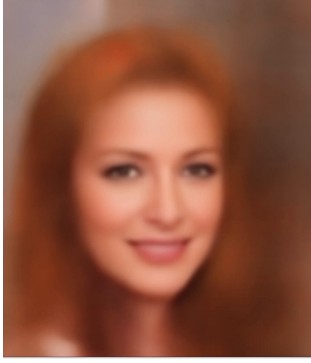

Figure 12: Interactive editing. Left: predicted image with a single-pixel manual coloured edit over the hair. Right: the mean of the calculated conditional distribution. N.B: The red circle highlighting the manual edit is for illustration purposes only and serves no computational purpose.

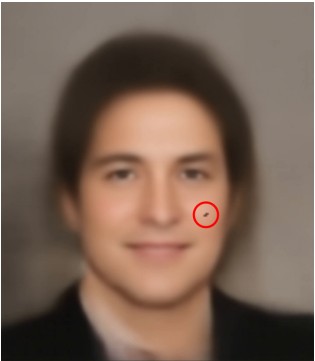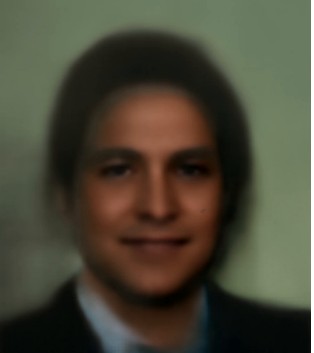

Figure 13: Interactive editing. Left: predicted image with a manual coloured edit over the skin. Right: the mean of the calculated conditional distribution. N.B: The red circle highlighting the manual edit is for illustration purposes only and serves no computational purpose.

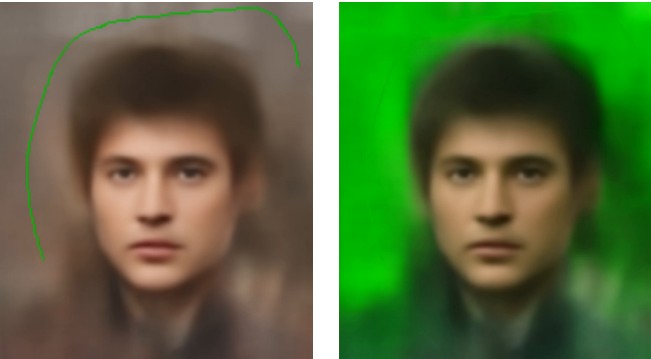

Figure 14: Interactive editing. Left: predicted image with a manual coloured edit over the background. Right: the mean of the calculated conditional distribution.

## D    Spherical Interpolation

For completeness, we include the spherical interpolation formula used in section 4.2.

$$\text{slerp}(\mathbf{a}, \mathbf{b}, t) = \frac{\sin((1-t)\omega)}{\sin \omega}\mathbf{a} + \frac{\sin(t\omega)}{\sin \omega}\mathbf{b} \tag{8}$$
$$\text{where} \quad \omega = \cos^{-1}(|\mathbf{a}| \cdot |\mathbf{b}|)$$

