# OpenReview forum: "Structured Uncertainty in the Observation Space of Variational Autoencoders"
_TMLR — Accepted by TMLR_

### Review · Reviewer_zquW · 2022-07-03

**Summary Of Contributions:**

This paper proposes an improvement in sample quality (spatial coherence) to the Variational Autoencoder (VAE) by incorporating pixel-wise correlations in the continuous output space. The authors do so by using a structured output distribution: they use a low-rank parameterization of a full covariance matrix in a multivariate Normal distribution. This is in contrast to the way in which most VAEs are parameterized in practice, where the observational distribution is pixel-wise independent. They demonstrate performance improvements on the CelebA and UK Biobank Brain Imaging datasets.


**Broader Impact Concerns:**

The authors included a sufficient section on ethical considerations (5.1).

**Requested Changes:**

Major:
- The biggest missing piece is a compelling demonstration that the structured output space in a VAE actually provides significant gains in sample quality. This can be demonstrated in a variety of ways. For example, the authors could apply their approach on top of existing models such as the NVAE, since this would lead to samples of similar quality. Or, the authors could report metrics that are used by existing methods (e.g. log-likelihoods) to demonstrate that they perform favorably.
- Such experiments or discussions would help clarify the take-home message. Is a structured output space complementary to existing approaches? Or is it enough to have the structured output space and ignore the other components? Those are the questions that still remain for me.

Minor changes:
- It'd be helpful to be self-contained and explain what it means to "pre-train" the mean (Mointeiro et al. 2020).
- Consistent notation: (nitpick) why not index z by i when talking about the ith data point x^{(i)} in Eq. 1?
- The prior p(z) should not have \theta in Eq. 2
- The true posterior p(z|x) should not be parameterized by theta (which parameterized the decoder, which is being learned) in Section 3.1
- Would the authors provide some guidance on how the hyperparameters were chosen? (e.g. e_H = -504750 doesn't seem like a very intuitive number to try as a first stab. Also, how should we go about choosing the rank of the covariance matrix relative to the dimensionality of the data, etc.?)


**Strengths And Weaknesses:**

Strengths:
- The paper is clearly written and easy to read overall.
- The authors clearly tried several approaches to make their approach work in practice, including proper initialization of NN weights, fixing the covariance diagonal, and incorporating an entropy constraint on the output distribution, etc.
- The authors provided a set of interesting experiments, such as interactive image editing.

Weaknesses:
- It's not very clear to me what the takeaway from this paper is. They seem to suggest that a structured observational distribution provides both computational (e.g. low-rank parameterization) and sample quality gains. But if that is the case, then it would have been nice to see comparisons of this method with Dorta et al. in terms of the number of model parameters or runtimes for the efficiency angle. If the main takeaway is improved spatial coherence of samples, then I have some more concerns below.
- The authors mentioned that their goal was not to obtain state-of-the-art sample qualities, but given the current submission it’s rather hard to tell: (a) how much their approach will help on larger models (that already produce good quality samples); (b) how much their approach will help on models with more complex structure in the VAE latent space; and (c) how well they are performing relative to other baselines in terms of commonly used metrics in the VAE community.
  - As written in the paper, the current literature has largely focused on architectural improvements and more complex models of the latent space. Such improvements (e.g. BIVA or NVAE) already produce spatially-coherent samples without additional considerations on the observational distribution. Given that the paper does not provide any way to compare across these approaches (either via models with similar architectures and different methods, or applying the authors’ method on top of these existing methods), it’s hard for me to believe that the structured observational distribution is actually providing significant gains when the baseline VAE model quality appears to be quite poor.
  - Even though log-likelihoods are not necessarily a good proxy for high sample quality (Theis et al. 2015), reporting these bpds allows for a fair comparison of this method across other baselines. It's hard to say how "good" an FID score of 100+ is on CelebA.



There were also some missing references that would be helpful to include:
- BIVA (Maaloe et al. 2019): architectural improvements to VAEs
- Correlated VAE (Tang et al. 2020): a similar idea to inducing correlation structures across dimensions, though this operates in the latent space of a VAE
- Bit prioritization (Shu et al. 2022)

---

> ### Author Response · Authors · 2022-07-26
> **Minor changes**
>
> 1 "It'd be helpful to be self-contained and explain what it means to "pre-train" the mean (Mointeiro et al. 2020)."
> Thank you for the suggestion. We have added clarification.
>
> 2 "Consistent notation: (nitpick) why not index z by i when talking about the ith data point x^{(i)} in Eq. 1?"
> In this instance, we are following the original notation from Kingma’s VAE paper. We believe both notations to be correct but opted for this choice.
>
> 3 "The prior p(z) should not have \theta in Eq. 2"
> Thanks for pointing this out. We have corrected the mistake.
>
> 4 "The true posterior p(z|x) should not be parameterized by theta (which parameterized the decoder, which is being learned) in Section 3.1"
> Again, thank you for catching this error we have addressed in the revised manuscript.
>
> 5 "Would the authors provide some guidance on how the hyperparameters were chosen? (e.g. e_H = -504750 doesn't seem like a very intuitive number to try as a first stab."
> The slack variables were chosen empirically by training an unconstrained model and observing the value that the entropy and Kl divergence take. The order of magnitude of these values was then used to inform the value of the slack variables/strength of the constraints. We have added a discussion on this in the revised manuscript.
>
> 6 "Also, how should we go about choosing the rank of the covariance matrix relative to the dimensionality of the data, etc.?)"
> The rank parameter is a trade-off between expressivity and computational cost. In practice, we found it sensible to select it based on the available computation resources, knowing that increasing it increases potential expressivity at the cost of extra resources.

---

> > ### Comment · Reviewer_zquW · 2022-07-29
> > **follow-up**
> >
> > Thanks for addressing the minor points. Including these recommendations and changes in the main text will be helpful.
> >
> > For the major changes:
> > 1) I agree that with respect to the improvements in sample quality from larger models such as NVAE, a big portion of it is probably with respect to complementary factors such as architectural design. However, my point was exactly that it's hard to tell how much of the gains we see from *this paper's approach* is due to the fact that the baseline VAE is pretty simple. If it is in fact true that these recent approaches fail to address a "different problem that persists" due to the "pixel-wise independent distribution, which remains to have disadvantages", it should be the case that adapting one of these existing approaches and adding on the pixel-wise dependence that this paper proposes should still lead to improvements. I think an experiment of this flavor is critical. It doesn't necessarily have to be the NVAE architecture (since it is quite expensive to train), but it should be a model that achieves comparable sample quality in terms of FID or ELBO to more up-to-date models (Beta-VAE is from 2016!).

---

> > > ### Author Response · Authors · 2022-07-29
> > > **On more modern VAE models**
> > >
> > > We do not have the resources to train NVAE from scratch. We would like to propose a compromise where we take a pre-trained NVAE and train only the last few layers, with and without the proposed low-rank likelihood distribution. Would this experiment address your concern?

---

> > > > ### Comment · Reviewer_zquW · 2022-08-07
> > > > **clarification on modern VAE model experiment and a bit more**
> > > >
> > > > On the additional experiment:
> > > > - Yes this modified experiment would help---I understand that training such a model from scratch would be infeasible. I just want to clarify that I am not too worried about whether or not the bigger VAE model has the exact same architecture as the NVAE (in fact, it could just be a variation of it). My primary concern is whether the gains from the proposed SOS-VAE is due to the fact that the baseline VAE is quite weak. This is shown via the very high FID numbers that are reported in the paper. So any additional experiment demonstrating that explicitly modeling the dependency structure between pixels in the observation distribution helps with sample quality---even with models that already give reasonable samples without this approach---would be convincing for me.
> > > >
> > > > Other minor things:
> > > > - There is an error in the paper (e.g. page 4, page 10, etc.) where pdfs are written as being drawn from certain distributions (e.g. $p_\theta(x \vert z) \sim \mathcal{N}(\mu,\Sigma)$). They should be corrected to: $p_\theta(x \vert z) = \mathcal{N}(\mu,\Sigma)$.
> > > > - There is an error in Figure 8's caption.

---

> ### Author Response · Authors · 2022-07-27
> **Response 'major'**
>
> Thank you for the suggestion to clarify the complementary nature of our approach compared to recent methods such as NVAE. We feel that an experimental comparison to NVAE would be less meaningful, precisely due to the orthogonality of the contributions. Hence, the sample quality and/or log-likelihoods would not be comparable.
>
> NVAE's output distribution indeed seems to produce samples that are sharp and of high quality. The distribution that is being used, however, is still pixel-wise independent (discretized logistic mixture), and cannot model spatially coherent samples in the observation space. The very nature of a pixel-wise independent distribution does not allow it to model spatial dependencies between pixels. Any variation to the mean must be uncorrelated, which can be defined as noise even if it is not visually apparent. The success of the NVAE's high quality samples largely stems from the hierarchical architectural design, using spatial latent variables and other design choices that result in spatial coherence, but not the pixel-wise independent distribution, which remains to have disadvantages that our paper addresses. Our work is complementary to the contributions in NVAE, addressing a different problem that persists in these recent approaches. Thus, our contributions, such as the ability to generate multiple plausible predictions and allowing efficient interactive editing with a single forward pass, stand independently to recent architectural contributions.
>
> On a practical note, training NVAEs is extremely resource demanding. On the GitHub repository (https://github.com/NVlabs/NVAE), the authors state “eight 16-GB V100 GPUs are used for training NVAE on CelebA 64. Training takes about 92 hours.” In contrast, our SOS-VAE is trained on a single 16-GB T4 GPU in 72 hours, which is a substantial difference in compute resources and of practical importance.
>
> In line with the request by reviewer 34NE, we have extended the discussion about state-of-the-art VAE architectures in the related work section. Please see the current revision for details.

---

### Review · Reviewer_34NE · 2022-07-09

**Summary Of Contributions:**

To use VAEs to do image generation, the authors improve the samples from the observational distribution. Specifically, Gaussian VAE’s decoder relies on a diagonal covariance matrix. The authors replace the diagonal covariance matrices with low-rank matrices (PP^T; see Section 3.2) plus diagonal covariance matrices. By doing so, the model can encode spatial dependencies among pixels better.

Moreover, the authors modify the VAE objective as shown in Eq. (1). The authors also try to minimize the entropy of the predicted distribution which can constrain the variance as a byproduct, as shown in Eq. (2).

Experiments show that samples from the proposed approach capture covariances among different parts of an image. The images shown in the paper look more spatially coherent.


**Broader Impact Concerns:**

The ethical consideration (written by the authors) is satisfactory.

**Requested Changes:**

- Add more information on tuning details of Eq. (2). How sensitive is the performance with respect to the slack variables and the Lagrangian multipliers? What would the authors suggest, if practitioners implement and experiment on the authors' approach?
- Discuss how well-tuned Dorta et al. (2018) is, given that it is an important baseline.
- Discuss whether the authors have tried adapting the above baseline onto multi-channel images.
- Quantify the effectiveness of Eq. (2) in terms of "reducing variance."
- Discuss FID scores more thoroughly.
- Discuss how the authors' approach is "compatible with state-of-the-art models" -- what is a state-of-the-art model; why will the authors' approach bring further improvement on such models?

**Strengths And Weaknesses:**

It is a good observation that Gaussian VAEs with diagonal matrices correspond to modeling each pixel independently.

While other work may be focusing on scaling up the models, this work models spatial dependencies (which is by modeling the covariance differently) more elegantly.

In general, the paper is well written and easy to follow.

---

I have a few questions on Eq. (2).

First, in Eq. (2), how are the hyperparameters tuned? For example, in CELEBA, the slack variables are 45 and -504750, respectively. How easy is it to obtain these numbers? It’s not clear how many runs are needed to get a satisfactory choice of the slack variables. It’s not clear how sensitive the results are with respect to the choice of slack variables. What should practitioners do in terms of hyperparameter tuning, if they want to use the authors’ method in the future?

Second, it is also not clear how the Lagrangian coefficients are determined.

Third, the motivation for using Eq. (2) is to better constrain the variance. It'll be great if there is a table/figure that empirically describes how much variance is reduced.

In terms of baseline, it’s great that the authors compared their performance with Dorta et al. (2018), which is designed for single-channel images. First, how well tuned are the hyperparameters / how well tuned is this baseline in general? Second, did the authors attempt adapting the method on multi-channel images? What difficulties are there?

In terms of evaluation metrics, FID scores can be introduced more thoroughly. Moreover, what does a difference of x points mean? How much of a difference shows that the authors’ approach is significantly better?

---

> ### Author Response · Authors · 2022-07-26
> **1 Add more information on tuning details of Eq. (2). How sensitive is the performance with respect to the slack variables and the Lagrangian multipliers? What would the authors suggest, if practitioners implement and experiment on the authors' approach?**
>
> The Lagrangian multipliers are not chosen - they are parameters learnt as part of training the model. Since the slack variables have a semantic value, they can be chosen empirically rather than through blind trial and error. As an example, the slack variable for the entropy constraint can be selected by observing the entropy of a predicted distribution from an unconstrained model. This observation can give the order of magnitude for the slack variable before further tuning. Training remains stable for slack variables in the correct order of magnitude. We have added words to this effect into the paper.

---

> > ### Comment · Reviewer_SxrT · 2022-07-27
> > **The inclusion of the slack variables as part of the optimized parameters was *not* at all apparent in the paper**
> >
> > I should've read this response before asking this question in my comments above. The inclusion of the slack variables among the tunable parameters of the model is not at all clear in the paper and should be explicitly mentioned.

---

> > > ### Author Response · Authors · 2022-07-27
> > > **Pleas see our reply to first comment**
> > >
> > > Thank you for the suggestion. We will include this in the next revision. Please see our reply to your above comment for further details.

---

> > ### Comment · Reviewer_34NE · 2022-08-07
> > **Resolved**
> >
> > I now see the additional clarification in the paper. Thanks!

---

> ### Author Response · Authors · 2022-07-26
> **2 Discuss how well-tuned Dorta et al. (2018) is, given that it is an important baseline.**
>
> We used our own implementation of Dorta et al.’s model predominantly using their original code, the same dataset and learning rate, trained for the same number of epochs. We have clarified this in our paper’s relevant section.

---

> > ### Comment · Reviewer_34NE · 2022-08-07
> > **Resolved**
> >
> > Thanks!

---

> ### Author Response · Authors · 2022-07-26
> **3 Discuss whether the authors have tried adapting the above baseline onto multi-channel images.**
>
> We did indeed attempt to modify Dorta et al.’s implementation to work with multi-channel images and also images of different resolutions. However, we, unfortunately, found working with their implementation extremely laborious as the dimensionality of the data was hard-coded and relied upon throughout the implementation. Furthermore, the authors did not supply a VAE implementation in their code, only an autoencoder. We, therefore, had to implement our own, using their supporting code, which locked us into the single channel and given resolution. Their code can be found here: https://github.com/Era-Dorta/tf_mvg

---

> > ### Comment · Reviewer_34NE · 2022-08-07
> > **More re: Dorta et al. (2018)**
> >
> > The comparison with Dorta et al. (2018) is quite important in my opinion, and I am glad that the authors did the comparison in the single-channel case.
> >
> > For Dorta et al.,
> > - If I'm understanding correctly, the implementation is the difficulty (especially because Dorta et al. didn't provide an VAE implementation).
> > - If there is an implementation, would the authors' argue that your methods prevail? Or how would the authors describe the pros and cons of your approach vs. Dorta et al., for multi-channel images (other than the existing paragraph at the end of Section 2)?

---

> > > ### Author Response · Authors · 2022-08-08
> > > **Response to 'More re: Dorta et al. (2018)'**
> > >
> > > Many thanks for appreciating the importance of our comparison with Dorta et al.'s method.
> > > - Yes, that is correct. Dorta et al. did not provide a VAE implementation, and importantly, the code that they did provide only supports single channel input images, as explicitly stated in their comments (please see https://github.com/Era-Dorta/tf_mvg/blob/01bc681a8b3aac5dcf0837d481b963f4968eb777/examples/autoencoder_mvg_chol_filters.py#L32)
> > > - Yes, we would argue that our method prevails. Besides the paragraph that you mentioned at the end of section 2, we would also like to highlight the paragraph at the end of section 4.1, where we discuss qualitative differences between the results of the two methods - we observed high-frequency distortions in the samples of Dorta et al. (2018) which are likely to be present also in multi-channel data (there is nothing suggesting that these artifacts are specific to the number of channels). We avoided making further claims that our method produces better samples qualitatively. However, we do use the FID metric additionally suggesting a quantifiable improvement of our method over Dorta et al. (2018).

---

> ### Author Response · Authors · 2022-07-26
> **4 Quantify the effectiveness of Eq. (2) in terms of "reducing variance."**
>
> The resulting average variance for each pixel across samples from the model trained without the entropy constraint is approximately 11.76, in comparison to approximately 0.0322 for the equivalently trained model with the entropy constraint and slack -504750. We have added these numerical results to the relevant appendix.

---

> > ### Comment · Reviewer_34NE · 2022-08-07
> > **Resolved**
> >
> > I see the updated caption for Figure 8.

---

> ### Author Response · Authors · 2022-07-26
> **5 Discuss FID scores more thoroughly.**
>
> What constitutes a significant reduction in the FID metric is not something we believe is openly discussed as quantifiable. We welcome examples to the contrary.
> However, we did generate 50,000 samples, as the authors of the FID metric do, from each model and propagated them, in turn, through the Inception-v3 model to sufficiently represent the set of synthesizable samples. This reduces sample bias to produce reliable results. We have added an explanation to the relevant section of the paper.

---

> > ### Comment · Reviewer_34NE · 2022-08-07
> > **Resolved**
> >
> > Resolved

---

> ### Author Response · Authors · 2022-07-26
> **6 Discuss how the authors' approach is "compatible with state-of-the-art models" -- what is a state-of-the-art model; why will the authors' approach bring further improvement on such models?**
>
> A recent architectural contribution is the NVAE (Vahdat & Kautz, A deep hierarchical variational autoencoder, NeurIPS 2020). NVAE's output distribution produces samples that are sharp and of high quality. The distribution that is being used, however, is still pixel-wise independent (discretized logistic mixture), and cannot model spatially coherent samples in the observation space. The very nature of a pixel-wise independent distribution does not allow it to model spatial dependencies between pixels. Any variation to the mean must be uncorrelated, which can be defined as noise even if it is not visually apparent. The success of the NVAE's high quality samples largely stems from the hierarchical architectural design, using spatial latent variables and other design choices that result in spatial coherence, but not the pixel-wise independent distribution, which remains to have disadvantages that our paper addresses. Our work is complementary to the contributions in NVAE, addressing a different problem that persists in these recent approaches. Thus, our contributions such as the ability to generate multiple plausible predictions and allowing efficient interactive editing with a single forward pass stand independently to recent architectural contributions.
> In line with the request by reviewer zquW, we have extended the discussion about state-of-the-art VAE architectures in the related work section.

---

### Review · Reviewer_SxrT · 2022-07-12

**Summary Of Contributions:**

This paper introduces an interesting alternative line of inquiry among the classes of VAE models by focusing on the representation of the observation distribution. An improved formulation of the predicted observation distribution is made by forming a full covariance matrix, removing standard assumptions of the generated pixel space being independent. To enable the execution of this approach, the covariance is estimated through a low-rank decomposition with additional adjustments to the optimization procedure to reduce instabilities while training. As a result of modeling a more complete approximation of the observation distribution, the generated images from the proposed VAE variant display greater spatial coherence and avoid uncorrelated pixel noise. Extensive qualitative analyses are performed to demonstrate the extensive capabilities of the Structured Observation Space VAE, including comparisons to related structure-inducing baselines.

**Broader Impact Concerns:**

I appreciate the effort taken in the writing of the current version of the paper that lays out some ethical considerations. Perhaps something that could be taken into consideration is that generative models based on human faces and other personally identifying features should be avoided altogether? A nice example of work that has done this is Creager, et al (ICML 2021, reference below) which develops a generative model over natural images (they use the waterbirds dataset for example). Would it be admissible to reconsider the use of CelebA entirely? Would the type of results demonstrated in this work be possible with a subset of ImageNet?



Creager, E., Jacobsen, J. H., & Zemel, R. (2021, July). Environment inference for invariant learning. In International Conference on Machine Learning (pp. 2189-2200). PMLR.

**Requested Changes:**

Forgive my boldness in naming the proposed method. I do think that so much of the paper and discussion of the results would be greatly improved if the new modeling approach had a name. This will make it easier for future work to clearly reference this paper and the insights it contains. We see this with the $\beta$-VAE. On top of the empirical improvements it provided, part of it's lasting impression is that it is recognizable by name. Please name your new model, feel free to take or improve upon my hastily determined SOSp-VAE if you like.

Please revise Section 3 following the recommendations given above.

Please run additional baselines that compare to the $\beta$-VAE and possibly a standard VAE that incorporates an entropy penalty. It's important to understand how the change to the objective function affects the comparison to the change of observation distribution. Without these ablations it is difficult to fully assess the impact of the separate improvements made with the SOSp-VAE. Is it possible that adjusting the objective as is done in Eqt 2, is sufficient to cover the outlined failings of standard VAE models? A comparison of both FID and qualitative features such as those done in Figure 2 between standard VAE, SOSp-VAE (done) as well as these two ablations would be really great.

On the note of Figure 2. It would be really helpful if each row was labeled on the left of the figure with the established notation in Section 3.

**Strengths And Weaknesses:**

This paper sets out a clear line of inquiry from its outset. I appreciated the focus on laying out the assumptions made among the various improvements to the standard VAE throughout the literature that highlight some limitations outsized limitations. This helps set-up the major areas of focus to guide the development of the proposed structured observation space VAE (if I may, SOSp-VAE):
1) Pixel-wise independence assumptions limit expressivity
2) Observation space is commonly modeled as an independent Normal distribution.
3) Generated samples from standard VAEs are not greatly differentiated from the mean of this indep. Normal dist.

The paper makes reasonable claims about the benefit of using a structured observation space, driven by allowing the optimization process to identify spatial correlations between pixels, thus improving the coherence of the generated images. An additional benefit that is possibly downplayed, is that a more complete representation of the data distribution is generated with a single forward pass of the model, enabling more efficient means by which to generate a diversity of samples. With the constructed mean and low-rank covariance, the SOSp-VAE is able to produce a flexible range of samples without needing additional expensive executions of the trained model. This is a very impressive finding and established result (among the breadth of additional insights provided in Section 4) that I believe the generative modeling community would find useful and interesting.

I am however a little surprised that the technical framing of the proposed SOSp-VAE is only covered lightly with a fairly vague section with limited formalization of the proposed modeling approach. As currently submitted, I cannot advocate for this paper to be published. There are far too few precise technical details about the proposed method. Much of the language surrounding the development of the covariance structure as well as the optimization procedure is far too vague and needs significant revision. I would recommend that the authors take the time to more fully and carefully outline their improvements to modeling the observation distribution by doing the following:
1) Clearly define the notation at the outset of Section 3 (eg. image example X of size S = HxW, etc)
2) Formalize the low-rank representation of the the covariance matrix, perhaps by more thoroughly describing Monteiro, et al (2020).

    a) As a side, note, is there a typo for the definition of the Diagonal matrix **D** in Section 3.2?
3) The initial description of the challenges of to the optimization of the improved obs. dist at the end of page 3 is great! This helps motivate the changes made to the objective function. But each of the adjustments made to the objective function should be clearly outlined. For example, moving to a $\beta$-VAE type objective is not at all mentioned. This should be better justified as well as the choice for the Lagrangian form of the objective. Just a little more precision and detail behind for the formulation would be necessary here.

A major omission in the technical introduction of Equation 2 is the lack of discussion around why multiple samples are needed in the first term and what M is.

More detail about the approach used in Dorta, et al (2018) is warranted. Following this, a more careful discussion that helps to differentiate this prior method from the proposed SOSp-VAE would be very helpful in understanding the technical grounding of the contributions.

Overall, the paper makes unsubstantiated claims or other statements about being closer to VAE theory in the development and use of the structured observation distribution. A more thorough development in Section 3 would help solidify the claims made throughout the paper on this front. Specifically, what parts of the theory informed the development and how does the SOSp-VAE satisfy this? Please be more precise and formal here.

I can understand the hesitation to expanding Section 3, but I believe it to be necessary to fully anchor the technical contributions made by the proposed SOSp-VAE. While the paper is good length, much of the space is taken by the figures and is clearly written. Of all the sections, the technical development in Section 3 is by far the weakest. Without more information and foundational development of the proposed method, I cannot envision the ML community really finding much use with this work despite the intriguing empirical results.

As an additional hesitation about the technical development of the SOSp-VAE, it's unclear how the low-rank covariance is actually represented. How are the parameters of the observation distribution actually produced by the model? How is the low-rank nature of **P** ensured?

I found Figure 2 to be a very neat way to validate the claims made about the failings of standard VAE models. I wonder if there's not a way to quantify the artifacts of this analysis (percentage of image different from mean, etc?).

The choices of the loss hyperparameters is opaque and seems random. More precision here outlining how the models were trained and what types of investigations were run to come to those parameters would be helpful. As currently submitted, the method and results are not at all reproducible. In this way, the paper seems incomplete.

---

> ### Author Response · Authors · 2022-07-26
> **1 Renaming the method**
>
> Many thanks for the naming suggestion. We agree that this makes it easier when referring to our modelling approach. We have adopted the suggested name throughout the manuscript.

---

> ### Author Response · Authors · 2022-07-26
> **2 Revision of Section 3.**
>
> We agree with the reviewer that section 3 needs to be elaborated and appreciate the helpful suggestions. We have extensively rewritten Section 3 and elaborated further on the details of our approach:
> (i) We have clarified the notation;
> (ii) We have elaborated further on the technical details of the method and formalised the representation of the covariance matrix;
> (iii) We have better justified the choices for the optimisation objective such as the use of Lagrangian optimization, and have clarified equation 2.
> (iv) We have further detailed the approach of Dorta et al. (2018) and how it differs from our method.
> Please see the updated pdf we will upload shortly

---

> > ### Comment · Reviewer_SxrT · 2022-07-27
> > **This will help, but what about the claims about being closer to VAE theory?**
> >
> > All of these points will help clarify the section but one important claim is not addressed here (or fully explained in the revised Section 3). In the paper, it is stated that SOS-VAE is closer to VAE theory. There was however no real substantive explanation of what this meant in the paper. I hope that the authors are taking this into account as they continue revise the paper.
> >
> > The updated Section 3 is much improved, thank you. **One additional question of major significance that re-surfaced as I read was how the rank $R$ is chosen in the paper (and for each experiment what was this value set to)?** _What guided the choice, was it simply another hyperparameter chosen alongside the Lagrangian slack variables?_
> >
> > As other reviewers have pointed out, and is currently unaddressed by the revisions, is how the specific Lagrangian variables were chosen for each experiment/dataset. Without clear details here, it would be difficult to implement the SOS-VAE on a new dataset and see similar results.

---

> > > ### Author Response · Authors · 2022-07-27
> > > **Rank and slack variable choice & VAE theory**
> > >
> > > The choice of rank has also been pointed out by reviewer zquW, where we replied with the following:
> > >
> > > "The rank parameter is a trade-off between expressivity and computational cost. As a hyper-parameter, the only unbiased way of selecting is through ablation. In practice, we found it sensible to select it based on the available computation resources."
> > >
> > > We will make sure that this is included in the next revision of the paper shortly.
> > >
> > > We see that you have found our reply to reviewer 34NE regarding the choice of slack variables and hope this has made clear how they are selected. Please note that we have already added this explanation to the current paper revision (penultimate paragraph of section 3). As you suggested, we will make explicit that the slack variables are tunable parameters in the next revision shortly.
> > >
> > > Regarding closeness to VAE theory, we will make this clear in the next revision as we agree it is an important clarification.

---

> > > > ### Author Response · Authors · 2022-07-27
> > > > **Requested revisions complete**
> > > >
> > > > We have added a discussion to section 3.3 in the latest revision (in orange text) regarding the choice of the rank hyperparameter. The penultimate paragraph in the same section now makes explicitly clear that the slack variables are tunable parameters.
> > > >
> > > > Our reasoning for our model more closely following VAE theory has been clarified in section 3.2 in the latest revision.
> > > >
> > > > We hope that you find these edits clear and informative.

---

> ### Author Response · Authors · 2022-07-26
> **3 Running additional baselines on entropy constraints**
>
> For the standard VAE (and beta-VAE) the entropy depends on the covariance matrix of the observational multivariate normal distribution. Commonly, practitioners use a fixed variance, meaning that the entropy is constant and therefore a penalty would have no effect. To compare with a “standard VAE” with an entropy penalty we must first introduce free parameters in the covariance matrix, either by: (i) using one free parameter for all diagonal values of the covariance (ii) or a free parameter per diagonal entry in the covariance matrix. Both these choices are unusual and would probably not be considered standard VAE practices.
> Regardless, in the case of a free diagonal, we observe, as expected, a reduction in variance. For a diagonal covariance, this means only a reduction in independent pixel noise, resulting in samples visually closer to the mean. However, there is still some noise in the sample, which can only be pixel-independent, therefore displaying none of the benefits our method provides.
> We will add the experiment with a free diagonal covariance in the appendix, including the FID scores as suggested.
> Note that the standard VAE in our experiments is the Beta VAE with the Lagrangian multiplier method and the same same slack variable for the KL loss. The only difference being the covariance matrix. This was done to ensure fair comparison.
>
> "Is it possible that adjusting the objective as is done in Eqt 2, is sufficient to cover the outlined failings of standard VAE models?"
> No, we believe not. Neither the entropy penalty nor the use of Lagrangian multipliers address the issue of the dependency between pixels. These two tools are used to stabilise training when using a low-rank covariance matrix. In addition, we used the Lagrangian multiplier method for the baseline as well and as previously mentioned the entropy penalty is of no consequence for with a fixed variance.
>
> "On the note of Figure 2. It would be really helpful if each row was labeled on the left of the figure with the established notation in Section 3."
> Thank you for the suggestion, we will incorporate it.

---

> > ### Comment · Reviewer_SxrT · 2022-07-27
> > **This justification and explanation should be included when introducing baselines**
> >
> > Thanks for the clarification. These types of details are necessary to include in the paper. Especially the use of the $\beta$-VAE as the baseline for the "Standard VAE". This is not at all clear in the paper and is misleading.
> >
> > The question about adjusting the objective in Eqt. 2 was pointing to whether a simple augmentation of the "standard" VAE loss could offer similar improvements to SOS-VAE. The authors have clarified why that wouldn't be sufficient to replicate the motivation behind SOS-VAE (spatial coherence). I however maintain that this could be an instructive baseline/ablation to perform to demonstrate the specific contributions of the modeling choices underlying SOS-VAE.

---

> > > ### Author Response · Authors · 2022-07-27
> > > **Requested ablation & Beta-VAE clarification**
> > >
> > > We agree and will make sure to clarify the use and choice of the Beta-VAE in the next revision shortly. Thank you for making this clear. As explained above, our motivation is for fairer testing by comparing models that differ only by our contributions.
> > >
> > > We agree that the requested ablation is valuable and will add it to the paper. Please note that this will take time to train the model to the standard of the paper.

---

> > > > ### Author Response · Authors · 2022-07-27
> > > > **Progress update**
> > > >
> > > > The clarification of the choice of Beta-VAE has been explicitly made in the latest revision of the paper, in orange text in section 4.1 and throughout. We hope this has resolved any misleading statements.
> > > >
> > > > The requested ablation is a work in progress.

---

> ### Author Response · Authors · 2022-07-26
> **4 Broader Impact**
>
> We agree with the reviewer that the use of human face data is problematic and ideally should be avoided. We appreciate the pointer to alternatives. At this stage, it would be difficult to replace all experiments due to the amount of work and compute resources that would be required. We have modified our statement for ethical considerations, and included a reference to the waterbird dataset.

---

> > ### Comment · Reviewer_SxrT · 2022-07-27
> > **Yeah, no reason to remove CelebA at this point, but...**
> >
> > I didn't intend for my comment to convey a suggestion to remove this dataset. I do however think that being upfront about the possible misuse of facial data is a very important consideration. I'm glad that this will be included in the updated submission.

---

> > > ### Author Response · Authors · 2022-07-27
> > > **Inclusion in revision**
> > >
> > > We agree with this suggestion - thank you. We have highlighted this in the current revision in section 5.1 and believe it makes for a more well-rounded section.

---

### Author Response · Authors · 2022-07-26
**New revision**

We have now uploaded a new version of the paper. Revisions can be identified by text in blue. We have incorporated the vast majority of recommendations left by the reviewers at this point and hope that the reviewers find these favourable. We welcome further discussion on both the revisions and our responses to reviewer comments in this forum.

---

### Author Response · Authors · 2022-07-27
**Another revision**

We have addressed further comments and a revision to the paper is now uploaded. The latest revisions can be identified by orange text, whilst previous revisions (to the original submission) remain in blue.

We hope that these further revisions are well-received and continue to welcome further discussion.

---

### Decision · Action_Editors · 2022-08-18

**Recommendation:** Accept with minor revision

**Comment:**

First — thanks to the authors for submitting this interesting work, thanks to the reviewers for your insight, and thanks to all for a productive discussion that appears to have clarified the manuscript and led to concrete proposals to improve the work.

To summarize the reviewer discussion — there was a consensus that the discussion, revisions, and discussed new experiments were satisfactory.

One reiterated concern is that the reported improvements are due to a relatively weak VAE baseline.  One reviewer explicitly suggested incorporating another experiment with 'a simpler dataset with a baseline that's known to produce good samples' as a way to understand the effect of the new approach.  Another author proposed an additional experiment with a stronger baseline.  The concrete experiment proposal from the authors — tuning a pre-trained NVAE model with and without the proposed low-rank likelihood distribution — would improve our understanding of the new approach, and will making the final version of this work more compelling.  We suggest this experiment be added to the next revision.

Thanks again for the submission!

---

> ### Author Response · Authors · 2022-09-05
> **Thank you to the reviewers and editor**
>
> Many thanks for everyone's time to assess and discuss our work. We are delighted that our manuscript has been accepted (subject to minor revision). We wanted to thank the reviewers and the editor again for the many helpful and constructive suggestions and for engaging with us in an active discussion throughout the review process.
>
> We are working on the remaining experiment and hope to have the final results for the camera-ready version ready very soon.

---

> ### Author Response · Authors · 2022-10-28
> **Camera-ready version**
>
> We apologize for the delay of submitting the camera-ready version. The final version is a clean copy of all revisions including the requested changes in addition to a more detailed discussion of recent deep VAE architectures. We would like to thank again the reviewers and the action editor for their patience and time in the review process. We felt very fortunate to have received many valuable and constructive suggestions for improving our work.

---

> ### Author Response · Authors · 2022-10-30
> **Code repository**
>
> Our code is now available here https://github.com/biomedia-mira/sos-vae